*Report*

# mitoTev-TALE: a monomeric DNA editing enzyme to reduce mutant mitochondrial DNA levels

Claudia V Pereira[1], Sandra R Bacman[1], Tania Arguello[1], Ugne Zekonyte[1], Sion L Williams[1], David R Edgell[2] & Carlos T Moraes[1],*

## Abstract

Pathogenic mitochondrial DNA (mtDNA) mutations often co-exist with wild-type molecules (mtDNA heteroplasmy). Phenotypes manifest when the percentage of mutant mtDNA is high (70–90%). Previously, our laboratory showed that mitochondria-targeted transcription activator-like effector nucleases (mitoTALENs) can eliminate mutant mtDNA from heteroplasmic cells. However, mito-TALENs are dimeric and relatively large, making it difficult to package their coding genes into viral vectors, limiting their clinical application. The smaller monomeric GIY-YIG homing nuclease from T4 phage (I-TevI) provides a potential alternative. We tested whether molecular hybrids (mitoTev-TALEs) could specifically bind and cleave mtDNA of patient-derived cybrids harboring different levels of the m.8344A>G mtDNA point mutation, associated with myoclonic epilepsy with ragged-red fibers (MERRF). We tested two mitoTev-TALE designs, one of which robustly shifted the mtDNA ratio toward the wild type. When this mitoTev-TALE was tested in a clone with high levels of the MERRF mutation (91% mutant), the shift in heteroplasmy resulted in an improvement of oxidative phosphorylation function. mitoTev-TALE provides an effective architecture for mtDNA editing that could facilitate therapeutic delivery of mtDNA editing enzymes to affected tissues.

**Keywords** heteroplasmy; I-TevI; mitochondrial DNA; mitoTev-TALE; monomeric

**Subject Categories** Genetics, Gene Therapy & Genetic Disease

## Introduction

Mitochondria contain their own double-stranded circular genome composed of 16,569 base pairs (bp). There are approximately one thousand copies of mitochondrial DNA (mtDNA) inside a cell. The mtDNA contains 37 genes, encoding 22 transfer RNAs (tRNAs), two ribosomal RNAs (12S and 16S), and 13 subunits of the oxidative phosphorylation system (OXPHOS; Anderson *et al*, 1981; Wallace, 2007). Mutations in the mtDNA have been associated with several clinical syndromes, which are in most cases maternally inherited and have a variable onset (Giles *et al*, 1980; Wallace, 2007; Christie *et al*, 2015). More than 270 point mutations and large-scale rearrangements in human mtDNA have been associated with diverse clinical phenotypes, including muscle weakness, cardiomyopathy, stroke-like episodes, optic neuropathies, and neurodegenerative disorders, among others (DiMauro & Moraes, 1993; Schon *et al*, 2012; Stewart & Chinnery, 2015). MtDNA mutations are commonly heteroplasmic, whereby wild-type and mutant genomes co-exist (Wallace & Fan, 2010; Wallace & Chalkia, 2013; Stewart & Chinnery, 2015) and the threshold levels for disease onset are usually between 70 and 90% mutant mtDNA (DiMauro & Moraes, 1993; Davis & Sue, 2011). Therefore, a small shift in mtDNA heteroplasmy should produce a significant improvement in patient's phenotype (Smith & Lightowlers, 2011). Because mitochondria lack an efficient double-strand break (DSB) repair system, mitochondria-specific endonucleases can lead to a quick degradation of the mutant mtDNA followed by a repopulation with wild-type (WT) mtDNA. This concept has been demonstrated by the use of different mitochondrial-targeted endonucleases including mitochondrial restriction endonucleases (REs), mitochondrial zinc-finger nucleases (mitoZFNs), and mitochondrial TAL effector nucleases (mitoTALENs) (Srivastava & Moraes, 2001; Tanaka *et al*, 2002; Minczuk *et al*, 2006, 2008; Bacman *et al*, 2007, 2010, 2013; Alexeyev *et al*, 2008; Gammage *et al*, 2014; Hashimoto *et al*, 2015; Reddy *et al*, 2015). Although these approaches could effectively shift heteroplasmy in cultured cells, they have limitations for gene therapy use. In the case of the REs, their usage is limited to rare mutations that create naturally occurring restriction sites; mitochondrial ZFNs and mitoTALENs, even though capable of targeting virtually any mitochondrial DNA sequence, work as dimers. TALENs are particularly large, which imposes restrictions in their packaging into viral vectors. The standard ZFN and TALEN architecture utilizes the dimeric *Fok*I Type IIS endonuclease domain (Bitinaite *et al*, 1998). Thus, both require two DNA recognition sites flanking a central spacer region.

Our group has previously described a dimeric mitoTALEN specific for the m.8344A>G MERRF mutation (Hashimoto *et al*, 2015). Here, we describe a strategy to overcome the architectural constraints imposed by mitoTALENs against the same target. Based on the GIY-YIG homing

1 Department of Neurology, University of Miami Miller School of Medicine, Miami, FL, USA
2 Department of Biochemistry, Schulich School of Medicine and Dentistry, University of Western Ontario, London, ON, Canada
 *Corresponding author. Tel: +1 305 243 5858; E-mail: cmoraes@med.miami.edu

nuclease I-TevI, we assembled a monomeric I-TevI-TALE nuclease (Kleinstiver *et al*, 2012, 2014; Beurdeley *et al*, 2013) targeting m.8344A>G, by taking advantage of the previously tested mitoTALEN-DNA binding domain. This novel design is a smaller alternative for mitochondrial genome editing, which is more readily applicable to *in vivo* delivery using adeno-associated virus (AAV) vectors.

# Results

### Designing mitoTev-TALEs against the m.8344A>G tRNA$^{Lys}$ mtDNA point mutation

We sought to design a mutant mtDNA-specific monomeric mitochondrial-targeted endonuclease. The homing nuclease I-TevI catalytic domain recognizes a short DNA sequence 5′CNNNG3′, thereby being relatively non-specific. However, I-TevI was reported to be effective in editing nuclear DNA as a monomeric fusion with a more specific TALE DNA binding domain (Beurdeley *et al*, 2013; Kleinstiver *et al*, 2014). Based on our previous TALE DNA binding domain targeting the MERRF m.8344A>G mtDNA (Hashimoto *et al*, 2015), we engineered two monomeric mitoTev-TALE nucleases, which differed in size by the TALE DNA binding domain repeated variable di-residues (RVD), either 8.5 or 11.5 RVDs. We also added a mitochondrial localization sequence (COX8/Su9) preceding the I-TevI catalytic domain, which was fused to the TALE DNA binding domain, through a flexible linker (depicted in Fig 1A). The fluorescent marker eGFP was placed downstream of the TALE domain, translated independently from the same transcript as the mitoTev-TALE due to a T2A picornavirus ribosome stuttering sequence (Fig EV1A), which also serves as an epitope tag for the nuclease (Bacman *et al*, 2013). The mitoTev-TALE was positioned to bind the anti-sense strand, where m.8344A>G contains a C (at position 3 of the binding domain, depicted in red in Fig 1B), which can be discriminated from the T present in the WT strand (Hashimoto *et al*, 2015). The I-TevI domain was positioned 12 bp apart from the TALE obligated T at position zero (T0, depicted in green in Fig 1B), which was the closest I-TevI required canonical sequence (5′CNNNG3′), in this case 5′CACTG3′ for binding and catalytic activity (Fig 1B).

Transient transfection in HEK293T and COS7 cells showed expression of full-length proteins (Fig 1C). As previously reported, the monomers showed more than one band with the T2A antibody, with the higher molecular weight band likely corresponding to a small fraction of the newly synthesized monomer with an uncleaved mitochondrial localization signal (Hashimoto *et al*, 2015). Both the 11.5 RVD and 8.5 RVD mitoTev-TALEs co-localized with the Mito-Tracker Red mitochondrial marker (Fig 1D).

### The 11.5 RVD mitoTev-TALE effectively shifted mtDNA heteroplasmy

The capacity of the monomeric mitoTev-TALE to shift mtDNA heteroplasmy was first tested in patient-derived transmitochondrial cybrids harboring approximately 40–45% of the MERRF mutation (MERRF Low Mut cells), previously characterized in our laboratory (Hashimoto *et al*, 2015). The cells were transfected with either plasmids coding the dimeric mitoTALEN (Hashimoto *et al*, 2015) or the monomeric mitoTev-TALEs. Cells were sorted for the respective

fluorescent markers, by mCherry plus eGFP (in the case of the dimeric mitoTALEN), or by eGFP only (Fig EV1). Sorted cell populations expressing both green and red after the mitoTALEN transfection were termed "Yellow", and cell populations sorted as negative for the fluorescent markers were termed "Black". The cells showing high green fluorescence after the mitoTev-TALE transfection were termed "GFP++", whereas non-fluorescent sorted cells were termed "GFP−+" (Fig EV1B). This nomenclature was chosen because of the consistent observation of low numbers of GFP-positive cells in the latter population, which increased overtime (Fig EV1C). Using PCR/RFLP, we found a shift in heteroplasmy after transfection with all three tested constructs. Although the shift was robust with the 11.5 RVD mitoTev-TALE, the shorter 8.5 RVD mitoTev-TALE produced a smaller shift (Fig 2A and B). Because of these initial results, we continued to use exclusively the 11.5 RVD mitoTev-TALE, hereafter referred to as "MERRF mitoTev-TALE". Our data showed a robust shift of 25% toward the WT mtDNA in the "GFP++" cells after transfection with the MERRF mitoTev-TALE, which was similar to the shift produced by the mitoTALEN (30%, Fig 2A and B). The "GFP−+" showed a small reduction in mutant mtDNA (8%) when compared to the untransfected cells. In addition, sorted cells were cultured for up to 15 days post-transfection and mtDNA heteroplasmy was re-analyzed (Fig 2C). The "GFP++" cells maintained the lower mutant load 15 days after sorting (Fig 2D).

### Heteroplasmic cybrids transfected with the MERRF mitoTev-TALE showed expression level-dependent reduction in mtDNA levels

Quantitative PCR analysis was performed in DNA samples from the sorted cybrids collected at two different time-points, 1 and 15 days after transfection. The data showed a decrease in the total mtDNA levels in the GFP++ cells, 1 day after transfection (Figs 2E and EV1D) but not in the GFP−+ cells. The lower levels of total mtDNA persisted in the GFP++ cells up to 15 days in culture (Figs 2E and EV1D). However, this decrease was not observed in WT cells 1 day after transfection which indicates that off-site cleavage by the MERRF mitoTev-TALE is relatively low, under the tested conditions (Fig 2F). The associated mtDNA depletion in the MERRF Low Mut cybrids is likely related to the elimination of the mutant mtDNA (Fig 2F).

### Changes in mutant load and total mtDNA levels in a cell clone with high levels of MERRF mutant mtDNA

In order to test the biological significance of the change in mtDNA heteroplasmy, we used a different cybrid clone with higher mutant load, with approximately 91% mutant mtDNA. This MERRF High Mut line (clone 20) was transfected with the MERRF mitoTev-TALE. Similar to what was described above, after 2–3 days, the cells were sorted for eGFP. A portion of the sorted cells was allowed to grow up to 27 days. A time-course experiment showed a shift from 8 to 23% WT mtDNA as early as 2–3 days after transfection (Fig 3A and B). The ratio of WT/Mut mtDNA was also analyzed at later time-points after sorting (15–20 days, 23–27 days) and increased to 43% WT mtDNA in the GFP++ population of cells, stabilizing over time (Fig 3B). In accordance with previous observations, the GFP−+ population of cells also showed a small shift in heteroplasmy toward the WT mtDNA which also increased significantly overtime (Fig 3B). The mtDNA levels were

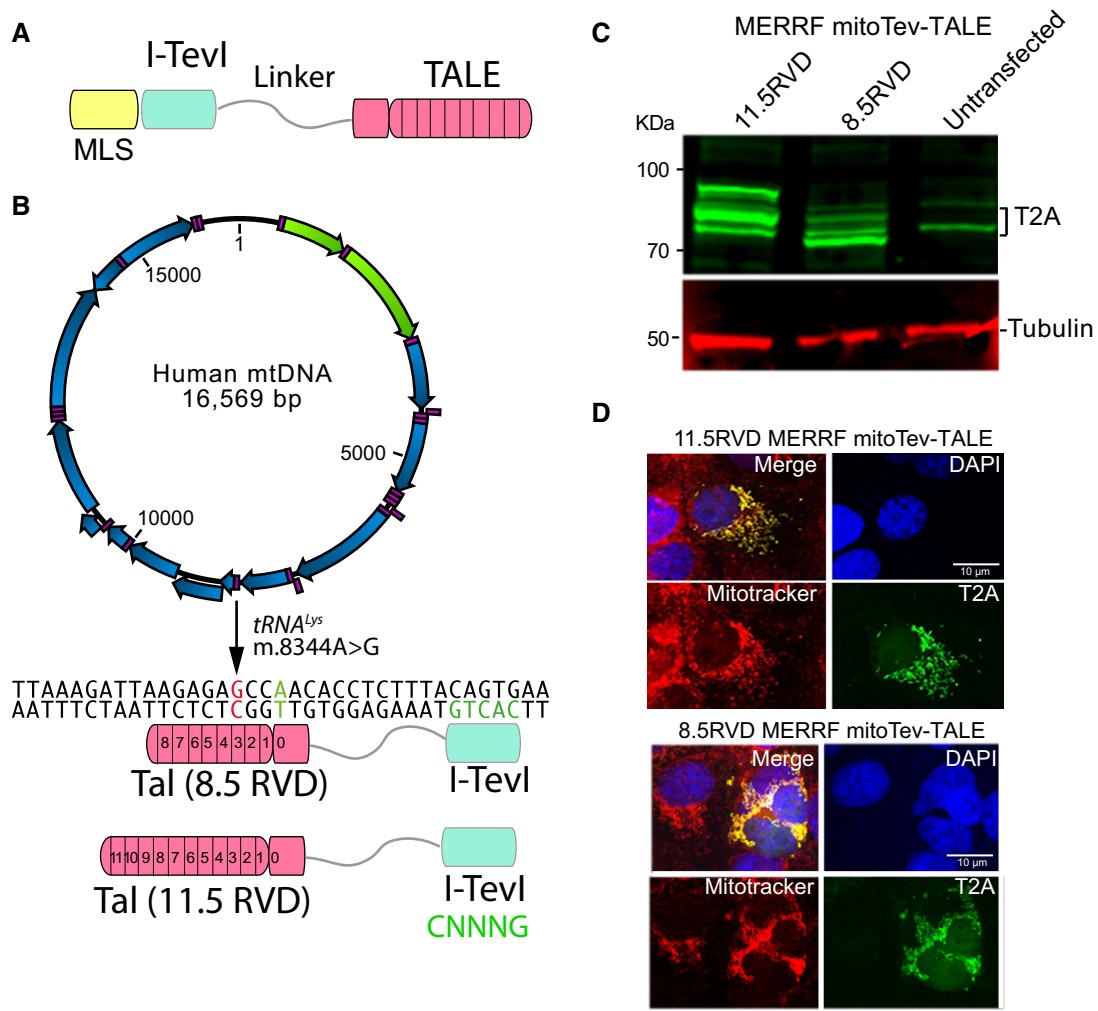

**Figure 1. Development of a mitoTev-TALE targeting the m.8344A>G mitochondrial DNA mutation.**

A   The structure of a mitoTev-TALE. It consists in a mitochondrial localization sequence (MLS) and a TALE DNA binding domain which is fused to a monomeric endonuclease (I-TevI), through a flexible linker (linker).

B   Diagram of mtDNA illustrating the binding site for two monomeric mitoTev-TALEs (8.5 and 11.5 RVDs) designed to target the m.8344A>G in the tRNA$^{Lys}$ gene. The two mitoTev-TALEs differ in the number of RVDs. The A:T base pair at position T0 is shown in green and the G:C discriminatory base pair in red. The I-TevI endonuclease domain targets a 5′CNNNG3′ sequence, in this case 5′CACTG3′ shown in green which is spaced by 12 bp from the TALE DNA binding site.

C   HEK293T cells were transfected for 48 h with each mitoTev-TALE (11.5 and 8.5 RVDs), and total cell extracts were used for Western blots with anti-T2A antibody. Molecular weights of the two different mitoTev-TALEs are ~85 kDa and ~70 kDa, respectively.

D   Immunocytochemistry using the same antibody as in (C) along with MitoTracker Red was performed in COS7 cells, 24 h after transfection. Nuclei were counterstained with DAPI (blue). 11.5 RVD and 8.5 RVD mitoTev-TALEs both co-localized with mitochondria, as seen in the merge image. Scale bar: 10 μm.

Source data are available online for this figure.

reduced in the GFP++ cells but not in the GFP−+ (Fig 3C). To better understand the correlation between MERRF mitoTev-TALE expression levels and mtDNA depletion, we conducted a narrower gating approach during FACS sorting. The high mutant cells were transfected for 48 h and gated/sorted by the intensity of GFP fluorescence. We aimed to separate the GFP−+ (low green) from the GFP++ (intermediate green) and GFP+++ (bright green)-positive cells (Fig 3D). We analyzed the total mtDNA levels in the different populations of sorted cells either at 2 days or at 15 days after sorting. As expected, the GFP−+ cells did not show a significant decrease in the total mtDNA levels, whereas the GFP++ had a partial reduction, and the GFP+++ population showed a higher

mtDNA depletion (Fig 3E). WT cybrids were analyzed in a similar manner and also showed decreased levels of total mtDNA levels in the GFP++ and GFP+++ cells, 2 days after transfection. However, the decrease in the mtDNA levels in these populations of WT cells was transient (Fig 3F).

**Cybrids with high mutant loads transfected with the MERRF mitoTev-TALE showed improved mitochondrial function**

To determine whether the treatment with mitoTev-TALE had phenotypic benefits to MERRF cells, we analyzed mitochondrial function in both GFP−+ and GFP++ cells. Cells transfected with the MERRF

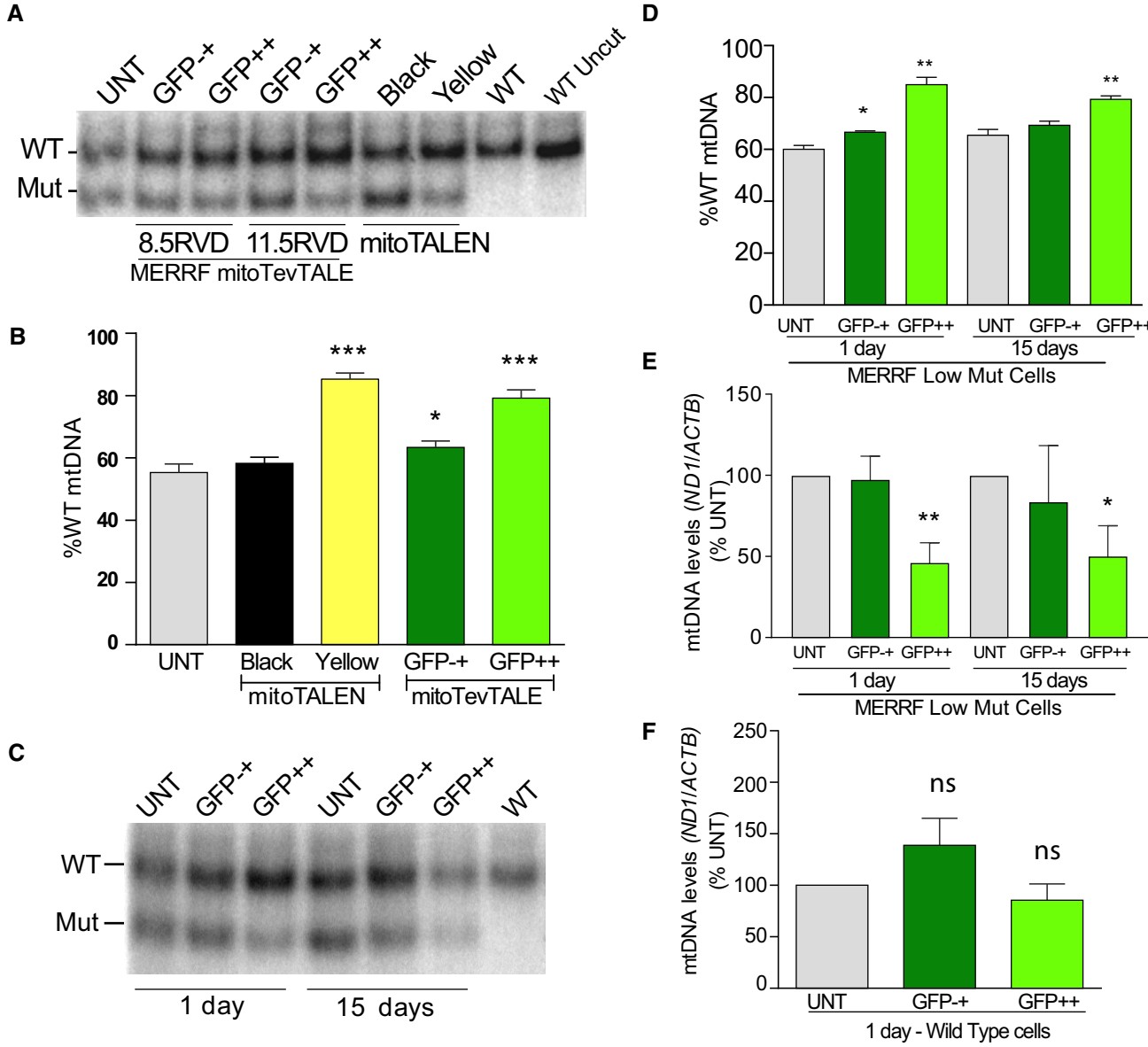

**Figure 2. Monomeric mitoTev-TALEs reduce mutant mtDNA loads in low mutant cybrids harboring the m.8344A>G mtDNA point mutation.**

A   MtDNA heteroplasmy analyzed by PCR/RFLP, 24 h after mitoTev-TALE and mitoTALEN transfection. The RFLP analysis shows increased %WT mtDNA in sorted cells when compared to the untransfected. mitoTALEN monomers positive for both eGFP and mCherry were isolated as "Yellow". The "Black" cells represent the mitoTALEN sorted population of cells negative for eGFP and mCherry, and the "GFP−+" population represents mitoTev-TALE sorted cells with low levels of eGFP fluorescence. GFP++ were cells positive for GFP after mitoTev-TALEs transfections.

B   Quantification of the % of WT load in cybrid sorted populations after transfection.

C   Representative RFLP analysis showing the change in heteroplasmy at 1 and 15 days, after sorting.

D   Quantification of the heteroplasmic shift over time.

E   Determination of the levels of mtDNA (*ND1/ACTB*) by qPCR of mutant cybrids transfected with MERRF mitoTev-TALE.

F   mtDNA levels in wild-type sorted cells transfected with the MERRF mitoTev-TALE. Data are expressed as percentage of the untransfected cells (%UNT).

Data information: Data are expressed as mean ± SEM of *n* = 5–12 independent experiments (B), *n* = 3 independent experiments (D), *n* = 3–6 independent experiments (E), and *n* = 6 independent experiments (F). Statistical analyses were performed by the use of unpaired two-tailed Student's *t*-test, *$P < 0.05$, **$P < 0.005$, ***$P < 0.001$. For exact *P*-values and *n* number, see Appendix Table S1.

Source data are available online for this figure.

mitoTev-TALE (both GFP−+ and GFP++) significantly increased oxygen consumption rates (OCR), when compared to the untransfected control (Fig 4A and B). Because it takes several weeks after sorting to obtain enough cells for functional analyses, the increase in

OCR in the GFP−+ population was not surprising as the small number of GFP-positive cells in this population likely grow faster than untransfected/defective cells. The results of several independent experiments showed a significant increase in all four bioenergetic parameters

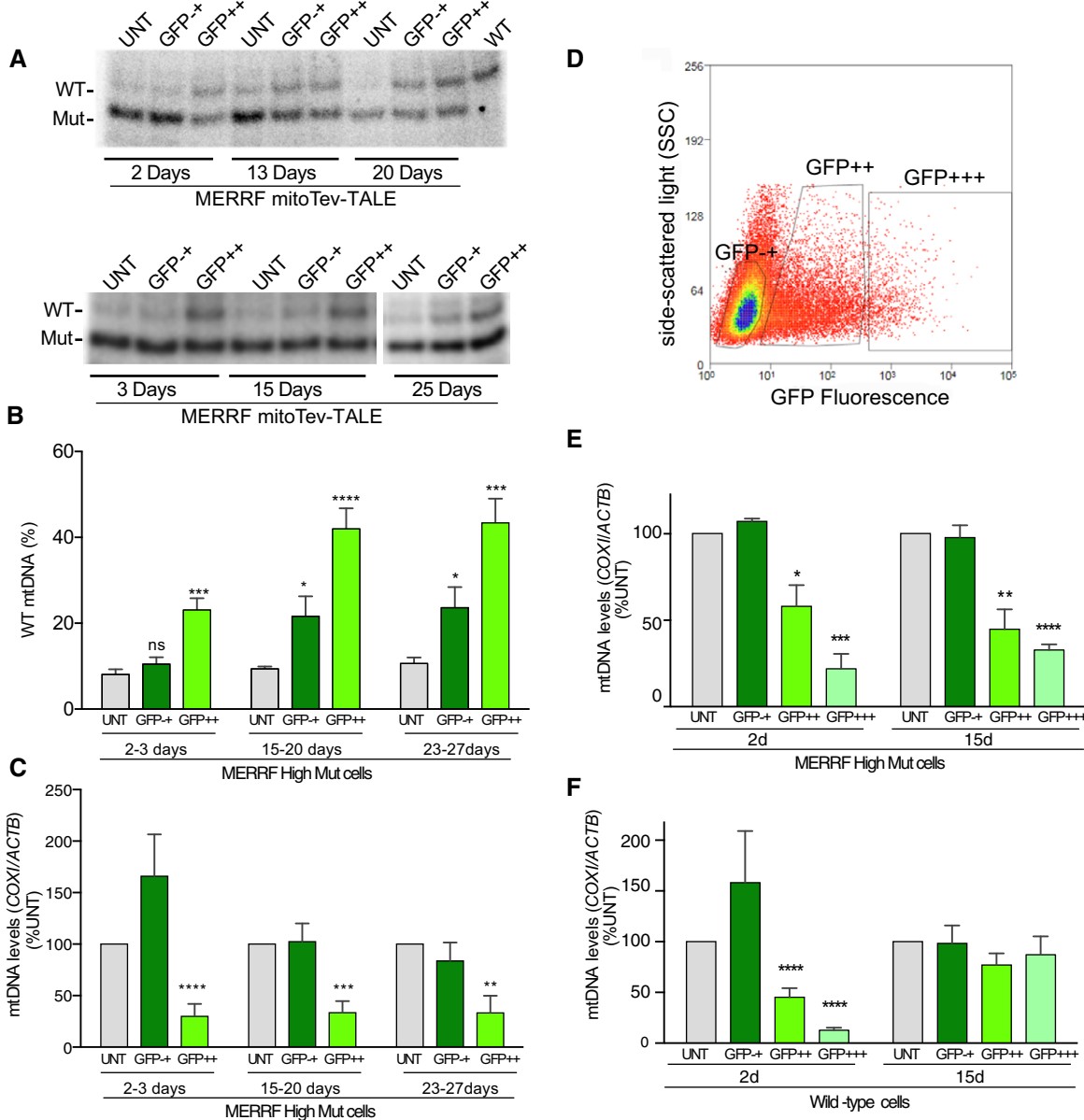

**Figure 3.  The MERRF mitoTev-TALE effectively shifts heteroplasmy in MERRF high mutant cybrids.**

A   RFLP analyses of high mutant cybrids transfected with the MERRF mitoTev-TALE. The sorted cells were analyzed at different times of growth by PCR/RFLP as previously described.

B   Quantification of WT loads of 6–8 independent experiments/sortings, analyzed at different time-points. Data are expressed as mean ± SEM.

C   Total mtDNA levels determined by qPCR in 6–7 separate sortings/experiments. Data are expressed as mean ± SEM.

D   Flow cytometry diagram of gated sorted cells. The GFP−+ represent the cells with low levels of fluorescence, GFP++ showed intermediate levels of fluorescence, while GFP+++ had the highest levels.

E   Quantification of mtDNA levels by PrimeTime qPCR probes. The mtDNA levels were analyzed in the different populations of sorted cells. Data are expressed as mean ± SEM of 3–4 separate experiments.

F   mtDNA levels were also determined in the WT cybrids as described above. Data are expressed as mean ± SEM of 4–7 separate experiments.

Data information: Statistical analyses were performed by a two-tailed Student's *t*-test; *$P < 0.05$, **$P < 0.01$, ***$P < 0.001$, ****$P < 0.0001$ vs UNT. For more details, see Appendix Table S1.

Source data are available online for this figure.

determined in the Seahorse apparatus (basal respiration, maximum respiratory capacity, spare capacity, and ATP-linked respiration) in both 27-day-grown GFP−+ and GFP++ cells when compared to the untransfected cells, which were grown in parallel (Fig 4B). The OCR in GFP++ cells was still lower than the WT cybrids (Fig 4B). Coupling efficiency was similar between the MERRF mitoTev-TALE-treated cells and the control WT cybrids, a bioenergetic parameter that was significantly decreased in the untransfected cells (Fig 4C).

    

The MERRF mutation in the tRNA lysine gene is known to cause impairment of mitochondrial translation by reducing tRNA lysine aminoacylation and mitochondrial protein synthesis (Enriquez *et al*, 1995; Yasukawa *et al*, 2000). To understand whether the increase in respiration in treated cells was accompanied by an increase in mitochondrial protein synthesis and OXPHOS protein levels, we analyzed mitochondrial translation. $^{35}$S-methionine labeling of cells in the presence of a cytoplasmic protein synthesis inhibitor showed a significant increase in the synthesis of mtDNA-encoded subunits in both GFP−+ and GFP++ populations (which were grown for 25–28 days after sorting to obtain enough cells; Fig 4D). In agreement, the levels of COXI (mtDNA-encoded complex IV subunit) and NDUFB8 (nuclear-encoded complex I subunit that is sensitive to unassembled complex I) showed increases in GFP−+ and GFP++ cells, although because of variability, some markers did not reach significance (Fig 4E). The nuclear-coded subunit of complex II, SDHA, did not change between the different cells, neither did tubulin (Fig 4E).

# Discussion

## The advantages of the mitoTev-TALE architecture

The manipulation of mtDNA heteroplasmy has great potential as a therapeutic tool. Although CRISPR/Cas9 has been reported to cleave mtDNA, a number of concerns suggest that it may be challenging to use CRISPR/Cas9 for mtDNA editing due to difficulties in nucleic acid import (Gammage *et al*, 2018). On the other hand, a broad range of DNA editing enzymes has been tested in mitochondria (Srivastava & Moraes, 2001; Tanaka *et al*, 2002; Bayona-Bafaluy *et al*, 2005; Bacman *et al*, 2007, 2010, 2012; Alexeyev *et al*, 2008), the most flexible being mitoTALEN and mitoZFN (Minczuk *et al*, 2008; Bacman *et al*, 2013; Gammage *et al*, 2014; Hashimoto *et al*, 2015). However, both architectures are dimeric, posing challenges to viral delivery due to limitations in construct size. The current study presents a

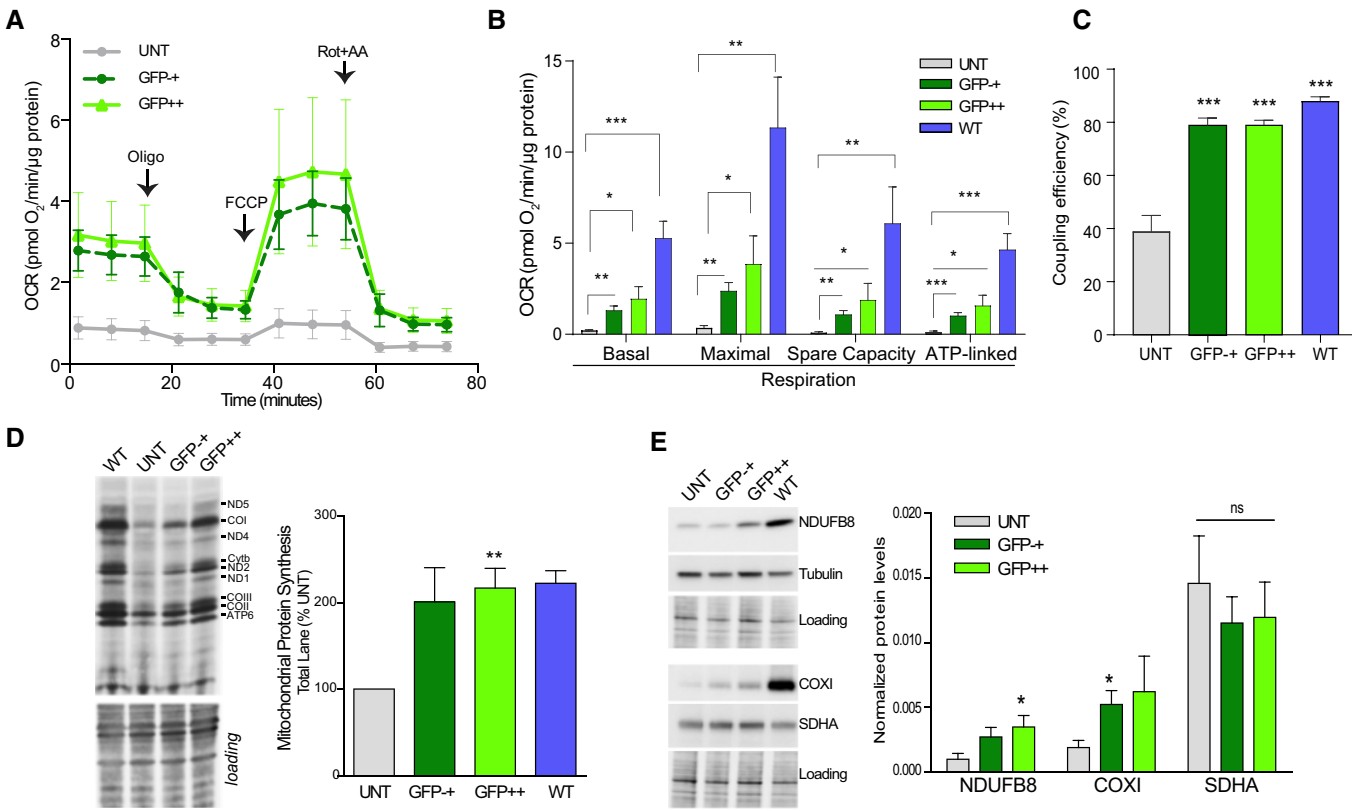

**Figure 4. The MERRF mitoTev-TALE significantly improves mitochondrial oxidative phosphorylation function of high mutant cybrids.**

A Oxygen consumption rate (OCR) upon sequential injection of oligomycin (Oligo), FCCP, and rotenone (Rot) + antimycin (AA), in untransfected, sorted "GFP−+" and "GFP++". Data represent the mean ± SEM of 5–7 separate experiments.

B Quantification of 4–6 independent experiments comparing the basal respiration, maximal respiration, spare respiratory capacity, and ATP-linked respiration of the UNT vs "GFP−+", "GFP++", and WT cells. The cells were analyzed between 25 and 28 days after sorting. Data represent the mean ± SEM.

C Coupling efficiency, measured as the ratio of ATP-linked/basal respiration × 100. Data represent the mean ± SEM of 4–6 separate experiments.

D Representative membrane of a mitochondrial protein synthesis experiment. The results were normalized to the gel loading. Graph represents five separate experiments (with exception of wild-type samples, n = 2). Data represent the mean ± SEM.

E Western blot showing the levels of NDUFB8 and COXI in untransfected and sorted cells. The SDHA and tubulin levels were also analyzed. The panel also shows the quantification of these markers normalized by the stain-free loading of six independent sortings/experiments. Data represent the mean ± SEM.

Data information: Statistical analysis was performed using two-tailed Student's *t*-test between each group pair (UNT vs GFP−+; UNT vs GFP++, and UNT vs WT), *P < 0.05, **P < 0.01, ***P < 0.001, ns, not significant. For exact *P*-values and *n* numbers, see Appendix Table S1.
Source data are available online for this figure.

promising alternative, which is more amenable to viral packaging and *in vivo* delivery.

Our laboratory has previously used mitoTALENs, which robustly target mtDNA and shift the heteroplasmy of cybrids harboring point mutations (m.14459G>A in *MT-ND6*, m.13513G>A in *MT-ND5*, and m.8344A>G in *MT-TK*) and the common deletion (m.8483_13459del4977; Bacman *et al*, 2013; Hashimoto *et al*, 2015). We have also explored reducing the DNA binding domain sequence in mitoTALEN monomers and demonstrated that monomers with 7.5–12.5 RVDs were able to discriminate single base differences (Hashimoto *et al*, 2015). Taking advantage of the success of the mitoTALEN approach, we now developed a different monomeric architecture that was initially developed for nuclear DNA editing (Kleinstiver *et al*, 2012; Beurdeley *et al*, 2013). We termed these enzymes "mitoTev-TALEs", which encompass the monomeric nuclease domain derived from the I-TevI homing endonuclease fused to a TALE DNA binding domain. This design can overcome the package limitations of bulky dimeric mitoTALENs, as the latter require approximately 7 kb per dimer, which are not well suited to viral vectors such as AAV that can only accommodate about 4.5–4.9 kb inserts. Of the two mitoTev-TALEs sizes tested, 8.5 RVDs and 11.5 RVDs, the longer one showed a more robust activity in changing heteroplasmy, and the size of this gene is approximately 2.5 kb (~ 3.2 kb with the CMV promoter). The fusion point of the I-TevI catalytic domain to the N-terminal TALE DNA binding domain influences the catalytic activity of the endonuclease, and the linker region between the two domains can also influence the specificity of the binding and the activity in the target DNA (Beurdeley *et al*, 2013; Kleinstiver *et al*, 2014). In this case, we did not change the fusion point and we kept the same I-TevI linker; thus, a possible explanation could be that the shorter version did not provide enough specificity for an appropriate binding to the target sequence and therefore decreased the overall efficiency of the heteroplasmy shift. Optimization of specificity can be further explored by a different TALE binding domain recognizing the opposite strand sequence fused to I-TevI mutants that recognize and cleave GNNNG (which is created by the MERRF m.8344A>G mutation). I-TevI mutants, such as the K26R, T95S, and Q158R mutations, recognize such sequence, instead of the canonical CNNNG (Roy *et al*, 2016).

### Balancing mtDNA mutant elimination and mtDNA depletion

Our study showed that the MERRF mitoTev-TALE could reduce the mutant mtDNA load by 15–20% after 2–3 days. The shift obtained after one cycle of transfection had a major impact on mitochondrial function. The MERRF mitoTev-TALE produced a similar increase in the WT/Mut ratios when compared to the dimeric mitoTALEN previously described (Hashimoto *et al*, 2015), which strongly supports the efficacy and potential usefulness of this compact architecture to target mutant mtDNAs. Heteroplasmy shifts were accompanied by mtDNA depletion, which has been previously observed by us (Bacman *et al*, 2013) and others (Minczuk *et al*, 2008) using mitoTALEN or mitoZFN, respectively. Depending on the expression levels, differential degrees of mtDNA depletion were observed in MERRF mutant cell clones but also in wild-type cybrids. However, in the case of the WT cybrids, the levels of mtDNA normalized faster than in the mutant cells, indicating that the mitoTev-TALE preferentially cleaves the target mutant mtDNA. Similarly, it was

reported that lower levels of mitoZFN expression were associated with milder mtDNA depletion and more robust heteroplasmy shift over time, as this lower level of nuclease expression caused less off-target cleavages (Gammage *et al*, 2016).

The significant mitochondrial function improvement of the high mutant cybrids after treatment confirmed that sharp threshold levels exist for WT/MERRF mutant mtDNA ratios for a defective biochemical phenotype (Larsson *et al*, 1992; Rossignol *et al*, 2003; Smith & Lightowlers, 2011). Our results indicate that it is possible to significantly improve overall mitochondrial function of cybrids harboring m.8344A>G by the expression of controlled levels of the MERRF mitoTev-TALE.

In conclusion, monomeric mitoTev-TALE architectures can be engineered to reduce the levels of MERRF m.8344A>G mutant mtDNA in heteroplasmic cells. Because of its monomeric nature, this novel tool can greatly facilitate future gene therapy attempts for mtDNA diseases.

## Materials and Methods

### mitoTev-TALE constructs design

The re-engineered constructs were made as synthetic fragments (gBlocks, Integrated DNA Technologies, Inc.) and cloned by using the In-Fusion HD cloning kit from Clontech Laboratories. The original plasmids were custom-made by Cellectis Bioresearch and modified as described in Bacman *et al* (2013). Monomeric mitoTev-TALEs were built by the removal of the FokI endonuclease domain from the original plasmid by digestion with restriction endonucleases (BamHI and PpUMI), removal of the Flag tag by PstI and EcoRV digestion followed by insertion of the I-TevI homing endonuclease catalytic domain composed of 137 amino acids (Tev137), followed by a small artificial linker after the mitochondrial localization signal (MLS), fused to the N-terminal of a previously characterized TALE DNA binding domain (Hashimoto *et al*, 2015). DNA binding domain was composed by either 8.5 or 11.5 RVDs targeting the m.8344A>G mtDNA point mutation, as depicted in Fig 1B. The dimeric mitoTALEN was previously designed according to Hashimoto *et al* (2015). All constructs were sequenced-verified.

### Cell culture transfections and sorting

Heteroplasmic cybrids harboring the m.8344A>G mtDNA, generated in our laboratory as previously described (Hashimoto *et al*, 2015), were transfected with 30 μg of plasmid using GenJet DNA *In Vitro* Transfection Reagent version II (SL100489, SignaGen Laboratories) in a T-75 flask, following the manufacturer's instructions. The cells were collected either 24 h, 48, or 72 h post-transfection for cell sorting analysis. Sorting was performed in a FACS Aria IIU by gating on single-cell fluorescence using a 561-nm laser and 600LP, 610/20 filter set for mCherry; and a 488-nm laser and 505LP, 530/30 filter set for eGFP. For some experiments, we also used a Beckman Coulter MoFlo Astrios EQ using a 100 μm nozzle at 25psi. Sorted cells were collected into separate tubes, one of them containing the sorted fluorescent cells either termed "GFP++" and "GFP+++" for some of the sortings, in the case of the mitoTev-TALEs; or "Yellow" when co-transfected with the two monomers containing mCherry and eGFP,

in the case of the mitoTALEN, as previously described (Hashimoto *et al*, 2015); we also isolated the untransfected sorted population of cells lacking fluorescence which were termed "Black" for the mito-TALENs and termed "GFP−+" for the mitoTev-TALE. The controls were also "sorted", in order to mimic the same conditions for the three or four cell populations.

## Immunocytochemistry analysis

COS7 cells were plated onto coverslips and transfected with each monomeric mitoTev-TALE (1 μg total, per well of a 6-well plate) for 24 h, as described above. In the following day, cells were stained with 200 nM MitoTracker Red CMXRos (M7512, Invitrogen, Waltham, MA) and incubated at 37°C for 1 h, protected from light. After, cells were fixed with 2% paraformaldehyde (PFA) for 15 min, at room temperature (RT). Cells were then permeabilized with a 0.2% Triton X-100 in phosphate saline buffer (PBS) during 2 min at room temperature, followed by blocking with 3% BSA in 1× PBS during 1 h at RT. Next, the primary antibody against T2A (Millipore ABS31, 1:200 in 3% BSA/PBS) was incubated for 1 h at RT. After washing the coverslips 3× with 1× PBS, the cells were incubated with the secondary antibody Alexa Fluor 488 goat anti-rabbit IgG (A-11008, Invitrogen 1:200 in 3% BSA/PBS) for 1 h at RT, protected from light. The coverslips were washed with 1× PBS and placed over a DAPI-containing mounting medium (VECTASHIELD HardSet Anti-fade Mounting Medium with DAPI, H-1500 Vector Laboratories), to stain the nucleus (Hashimoto *et al*, 2015). Images were recorded in a Zeiss LSM510 confocal microscope.

## mitoTev-TALE expression and OXPHOS protein levels assessment by Western blot

HEK293T cells were collected 48 h after transfection and total proteins were extracted using RIPA buffer, plus protease inhibitors (cOmplete™ Mini Protease Inhibitor Cocktail, Roche). Protein quantification was determined with the DC protein assay (5000116, Bio-Rad), according to the manufacturer's instructions. Total protein (30 μg) was subjected to electrophoresis in 10% Mini-PROTEAN® TGX™ Precast Protein Gels (4561034; Bio-Rad) and transferred to a nitrocellulose membrane (162-0115, Bio-Rad). The T2A antibody [Millipore ABS31, 1:1,000 in 1% blocking buffer for fluorescent Western blotting (MB-070, Rockland)] and α-tubulin (T9026; Sigma-Aldrich; 1:5,000 in 1% blocking buffer) were incubated overnight, under constant shaking, at 4°C. Secondary antibodies, anti-mouse IgG (goat) antibody DyLight™ 680 conjugated (610-144-002 Rockland; 1:5,000 in 1% blocking buffer) for tubulin and anti-rabbit IgG (goat) antibody DyLight™ 800 conjugated (611-145-122 Rockland, 1:5,000 in 1% blocking buffer) for T2A, were incubated for 2 h, at RT. After washing 3× with PBS, the membrane was scanned in an Odyssey Infrared imaging system (LI-COR, Lincoln, NE). The protein sequence of the constructs is shown in Fig EV2.

Proteins from transfected cybrids were also analyzed by Western blot using different primary antibodies such as NDUFB8 (ab110242; Abcam, 1:1,000 dilution), COXI (ab14705; Abcam, 1:1,000 dilution), SDHA (ab14715; Abcam, 1:1,000 dilution), and α-tubulin (T9026; Sigma-Aldrich, 1:5,000 dilution). These western membranes were developed with SuperSignal™ West Pico Chemiluminescent Substrate

(34080, Thermo Scientific), and images were obtained with a Chemidoc Imaging System (Bio-RAD). The use of TGX Stain-Free™ Precast Gels allowed the determination of protein loading through gel activation with the imager, following manufacturer's instructions. The protein levels were quantified with the Image laboratory software (Bio-RAD) and normalized to the quantification of stain-free loading.

## DNA extraction and quantification of m.8344A>G by "last-cycle hot" PCR and RFLP

Total genomic DNA was extracted from the sorted cells with the NucleoSpin Tissue XS kit (740901.50; Macherey-Nagel, Clontech, Mountain View, CA) according to the manufacturer's instructions. The DNA concentration was determined spectrophotometrically in a plate reader (BioTek Synergy H1 Hybrid), and samples were diluted for further analysis by PCR followed by a "last-cycle hot" PCR (Bacman *et al*, 2013; Hashimoto *et al*, 2015). As previously reported (Hashimoto *et al*, 2015), the mutant allele for the m.8344A>G mutation completes a mismatched primer with *Bgl*I half-site after PCR amplification which creates a restriction fragment length polymorphism (RFLP).

The following primers were used: F-CTACCCCCTCTAGAGC CCAC; R-GTAGTATTTTAGTTTGGGGCATTTCACTGTAAAGCCGTG TTGG (Hashimoto *et al*, 2015). Therefore, PCR products were digested with *Bgl*I (R0143S, New England Biolabs) in NEBuffer 3.1 (B7203S, New England Biolabs), for 4 h at 37°C (Hashimoto *et al*, 2015). An alternative strategy was performed to demonstrate completion of digestion of the WT mtDNA. Untransfected mutant cells and respective sorted populations (GFP−+ and GFP++) were evaluated with the alternative RFLP strategy. The same reverse primer was used but a new forward primer was designed (F2- CCC CTA AAA ATC TTT GAA ATA GGG CCC GTA TTT ACC CTA TAG CCC ACC GGG CCTAC) which creates an extra *Bgl*I site in both wild-type and mutant mtDNA sequences (Fig EV3).

For both RFLP designs, digested products were resolved in 12% polyacrylamide gels and WT samples were included as controls (digested and undigested). Radioactive signal was quantified using a Cyclone phosphorimaging system (PerkinElmer) and OptiQuant software (PerkinElmer, Waltham, MA).

## Total mitochondrial DNA levels determination by quantitative PCR (qPCR)

Quantitative PCRs using SYBR/ROX chemistry (172–5,264; Bio-Rad, SsoAdvanced SYBR Green) on a Bio-Rad CFX96/C1000 qPCR machine were performed, and the $\Delta\Delta C_T$ values were calculated using manufacturers' software (Bacman *et al*, 2013). We determined the levels of mtDNA with the following primers: *ND1* (F-CGA TTC CGC TAC GAC CAA C T; R-AGG TTT GAG GGG GAA TGC TG) and the nuclear DNA-encoded β-actin (*ACTB*) primers were used (F-GCG CAA GTA CTC CGT GTG GA; R-GCG CAA GTA CTC CGT GTG GA, located in exon 6) to normalize the results (Bacman *et al*, 2013). Alternatively, and to confirm the SYBR results, the qPCR was also performed with PrimeTime qPCR probes (IDT, Integrated DNA Technologies) according to the manufacturers' protocol (IDT) for both mtDNA (*COXI*) and nuclear DNA (*ACTB*). The probe sequence for *ACTB* is proprietary (ref. Hs.PT.56a.41086380.g from IDT), but

the primers sequence used were F- GCC ATG TAC GTT GCT ATC CA and R- GTC ACC GGA GTC CAT CAC). For *COXI,* the primers/probe sequences were as follows: F-TTC TGA CTC TTA CCT CCC TCT C; R-TGG GAG TAG TTC CCT GCT AA; probe:/56-FAM/TC CTA CTC C/ZEN/T GCT CGC ATC TGC TA/3IABkFQ/. The results were expressed as percentage of the untransfected cells (%UNT).

**Oxygen consumption rate (OCR)**

Oxygen consumption was measured at 37°C using a Seahorse XFp Extracellular Flux Analyzer (Seahorse Bioscience, Billerica, MA). The cells were seeded at a density of 20,000 cells/100 μl/well into six different wells of an XFp cell culture plate (wells A and H contained media only). In addition, an XFp sensor cartridge for each cell plate was placed in a 8-well calibration plate containing 200 μl/well calibration buffer and left to hydrate overnight at 37°C. The cell culture medium from the cell plates was replaced in the following day with 175 μl/well of pre-warmed low-buffered Seahorse medium supplemented with 10 mM glucose, 1 mM pyruvate, and 2 mM glutamine followed by incubation at 37°C for at least 1 h to allow the temperature and pH of the medium to reach equilibrium before the first-rate measurement. Optimal cell seeding density was previously determined. Measurements of endogenous respiration were followed by 1 μM of oligomycin, 0.5 μM of FCCP, and 1 μM of rotenone plus antimycin A. All oxygen consumption rates and bioenergetic parameters were determined as indicated in the excel report generators from Wave Software. The results were normalized to μg of protein per well quantified after each Seahorse run, with the DC protein assay, as described above.

**Mitochondrial protein synthesis analysis**

Cells were grown at 90% confluency in 6-well plates, prior to the experiment. Mitochondrial protein synthesis was performed in L-Methionine-L-cysteine-free DMEM 1× Media (Gibco) supplemented with 10% dialyzed FBS (Sigma, St. Louis, MO) and sodium pyruvate (1 mM) (Gibco). Cells were incubated for 10 min with the cytoplasmic inhibitor Emetine (Sigma, St. Louis, MO) at 100 μg/ml as an irreversible cytoplasmic protein inhibitor. Approximately 300 μCi/ml of Express [35S] Methionine-Cysteine protein labeling mix (PerkinElmer Life Sciences) was added to the media, and the radioisotope pulse labeling was performed for 30 min. After that, the media was replaced with DMEM Media, supplemented with 10% FBS, and 1 mM sodium pyruvate for a short incubation of 5 min prior to cell collection. Protein extraction was performed using RIPA buffer, and quantification was performed using the DC Protein Assay Kit (Bio-Rad). Approximately 50–70 μg were loaded into SDS polyacrylamide (30% acrylamide and 0.2% bis-acrylamide) gradient (17.5% gel, 5% stacking) gels. The running buffer 1× was adjusted to pH 8.3, and the gel was run at 100 V for 15–17 h. Transfer was performed to nitrocellulose membranes (Bio-Rad) using the Owl™ HEP Series Semidry Electro Blotting System for at least 3 h. The membrane was dried and exposed to film, and quantification of bands was done using the Image laboratory software (Bio-RAD). Fast Green FCF (0.002%) staining was used as total protein loading control. The total mitochondrial protein synthesis was determined by the quantification of the entire lane

**The paper explained**

**Problem**

Currently, there is no effective treatment or cure for mitochondrial diseases caused by heteroplasmic (mix of mutant and wild-type) mtDNA mutations. When a pathogenic point mutation reaches a certain threshold (above 70–90%), mitochondrial disease can manifest. The current mtDNA gene editing tools such as mitoTALENs showed to be effective to reduce the mutant mtDNA load in patient-derived cells. However, due to their large size and dimeric architecture, their packaging into viral systems is only possible with dual viral preparations, affecting *in vivo* delivery.

**Results**

Here, we report an alternative monomeric architecture to shift mitochondrial DNA heteroplasmy in patient-derived cells harboring different levels of the m.8344A>G mtDNA point mutation, associated with myoclonic epilepsy with ragged-red fibers (MERRF). We showed that a mitoTev-TALE, composed of a TALE DNA binding domain and a I-TevI catalytic domain, effectively shifts heteroplasmy in the tested cells. Furthermore, the shift was accompanied by an overall improvement in mitochondrial function.

**Impact**

The effective mutant elimination, as well as the monomeric nature of our construct, can facilitate delivery into affected tissues, enhancing the therapeutic potential.

normalized to the loading control lane, for each tested condition. The results were expressed as %UNT cells.

**Statistical analysis**

Data analysis was performed using GraphPad Prism 7 (La Jolla, CA) and presented as mean ± SEM. Statistical analysis of each group pair (UNT vs GFP−+, UNT vs GFP++, UNT vs GFP+++, or UNT vs WT) was performed by unpaired two-tailed Student's *t*-test. A *P*-value of < 0.05 was considered significant. The exact *P*-values and *n* numbers used are described in Appendix Table S1.

**Expanded View** for this article is available online.

## Acknowledgements

We would like to acknowledge the assistance of the Sylvester Comprehensive Cancer Center's Flow Cytometry core for the cell sorting services, namely to Patricia Guevara for her valuable help. We would also like to thank Dr. Oliver Umland from the Diabetes Research Institute Flow Core Facility at University of Miami Miller School of Medicine for assistance during cell sorting. This work was supported primarily by the Muscular Dystrophy Association, American Heart Association (15POST22430003) and National Institutes of Health (5R01EY0108041), and a Discovery Grant (RGPIN-2015-04800) from the Natural Sciences and Engineering Research Council of Canada to D.R.E. The following grants also provided secondary support (R01AG036871, 1R01NS079965) and the United Mitochondrial Disease Foundation.

## Author contributions

CVP performed all the experiments and planned the project with CTM. TA performed the mitochondrial protein synthesis experiments. SRB provided

mutant cybrids and the mitoTALENs that were used as a positive control. UZ helped with production of high mutant cells. CTM, SLW, and SRB designed the original constructs (mitoTALENs) which were modified for this study. DRE helped with reagents and insights during research progress. All authors contributed to the writing and editing of the manuscript.

## Conflict of interest

The authors declare that they have no conflict of interest.

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
