## [Review Process File · EMBO Molecular Medicine]

MitoTev-TALE: a monomeric DNA editing enzyme to reduce mutant mitochondrial DNA levels

Claudia V. Pereira, Sandra R. Bacman, Tania Arguello, Ugne Zekonyte, Sion L. Williams, David R. Edgell and Carlos T. Moraes

Review timeline:

Submission date:	27 May 2017
Editorial Decision:	26 June 2017
Revision received:	17 May 2018
Editorial Decision:	15 June 2018
Revision received:	18 June 2018
Accepted:	21 June 2018

Editor: Roberto Buccione/Céline Carret

Transaction Report:

1st Editorial Decision

26 June 2017

Thank you for the submission of your manuscript to EMBO Molecular Medicine and apologies for the delay in providing you with a decision. We experienced difficulties in securing three willing and appropriate reviewers and furthermore, their evaluations were delivered with some delay.

We have now heard back from the three expert Reviewers whom we asked to evaluate your manuscript.

As you will see, all three Reviewers are positive but raise a number of serious concerns. I will not go into much detail, as the comments are detailed and self-explanatory.

As you will see there is a high degree of convergence among the reviewers. In brief, the main concerns include: 1) lack of consistency in the time points used for the analyses; 2) need to make a better case as to whether the new approach represents a valid alternative to mitoTALENs and 3) unclear demonstration of rescue and poor execution of the Seahorse experiments. They also list a number of other issues, including of a technical nature and the need to improve the discussion and contextualization of the findings.

In conclusion, while publication of the paper cannot be considered at this stage, given the potential interest of your findings and after reviewer cross-commenting and internal discussion, we have decided to give you the opportunity to address the criticisms.

We are thus prepared to consider a substantially revised submission, with the understanding that the Reviewers' concerns must be addressed with additional experimental data where appropriate and that acceptance of the manuscript will entail a second round of review. The overall aim is to significantly upgrade translational relevance, conclusiveness, and the quality of data presentation. I understand that if you do not have the required data available at least in part, to address the above, this might entail a significant amount of time, additional work and experimentation and might be technically challenging, I would therefore understand if you chose to rather seek publication

elsewhere at this stage. Should you do decide to do so, and we hope not, we would welcome a message to this effect.

Please note that it is EMBO Molecular Medicine policy to allow a single round of revision only and that, therefore, acceptance or rejection of the manuscript will depend on the completeness of your responses included in the next, final version of the manuscript.

EMBO Molecular Medicine now requires a complete author checklist (<http://embomolmed.embopress.org/authorguide#editorial3>) to be submitted with all revised manuscripts. Provision of the author checklist is mandatory at revision stage; the checklist is designed to enhance and standardize reporting of key information in research papers and to support reanalysis and repetition of experiments by the community. The list covers key information for figure panels and captions and focuses on statistics, the reporting of reagents, animal models and human subject-derived data, as well as guidance to optimise data accessibility. This checklist especially relevant in this case given the issues raised with respect to statistical treatment and animal numbers.

As you know, EMBO Molecular Medicine has a "scooping protection" policy, whereby similar findings that are published by others during review or revision are not a criterion for rejection. However, I do ask you to get in touch with us after three months if you have not completed your revision, to update us on the status. Please also contact us as soon as possible if similar work is published elsewhere.

We now mandate that all corresponding authors list an ORCID digital identifier. You may acquire one through our web platform upon submission and the procedure takes <90 seconds to complete. We also encourage co-authors to supply an ORCID identifier, which will be linked to their name for unambiguous name identification.

I also suggest that you carefully adhere to our guidelines for publication in your next version, including our new requirements for supplemental data (see also below) to speed up the pre-acceptance process in case of a positive outcome.

***** Reviewer's comments *****

Referee #1 (Remarks):

Heteroplasmic mutations of mtDNA are common cause of mitochondrial diseases. Affected organs manifest the biochemical defect only when the proportion between mutated and wild type mtDNA exceeds a critical threshold. Recently, different approaches have been tried to reduce the copies of mutant mtDNA. For instance, Moraes' group developed a method to localize TALENs to mitochondria and cleave different classes of pathogenic mtDNA mutations leading to reduction in mtDNA heteroplasmy in patient-derived cells (1). The use of this approach has been suggested as potential therapeutic tool for improving a patient's phenotype. Although promising, the system presents a crucial limitation in term of size of the enzymes (TALENs are dimers and large) that would make complicate to package their coding regions into a viral vector. In this manuscript the authors presented an attempt to facilitate the therapeutic delivery of mitochondrial editing enzyme in humans. Indeed, Pereira and colleagues described the new engineered monomeric mito Tev-TALE, as innovative strategy able to overcome the limitations in the architecture of mitoTALENs. As proof of concept, they showed that mito Tev-TALE can efficiently shift the mtDNA ratio towards WT copy in patient-derived cybrids harboring m.8344A>G mtDNA mutation.

Comments

The manuscript is well structured and shows a clear novelty in the field. However, I have many regarding the quality of the presented data. Therefore, I recommend a major revision.

- One of the main issues of this work resides on complete lack of consistency in the time points used for the analyses. For instance, cells are often analyzed after 1-2 days and between 15 and 23 days after FACS sorting. The second time point, described as 15* days, is actually a period of 8 days that most likely would explain the huge variations observed in many experiments. The authors should

explain how and why they decided these time points and why these also differ between cybrids with 45% and 91% of mutation load. The experiments need to be performed again at 15 and 25 days.

- In Figure 2 and Figure S4, quantification of mtDNA levels after transfection with mito Tev-TALE were shown. The results seem to be not clear and convincing:

- In most of the experiments mtDNA-coded ND1 and CYTB were used as probes to quantify mtDNA (normalized for ACTB). I would suggest to plot these data on the same graph. At 1 day after transfection, the observed mtDNA depletion with ND1 as probe was quite severe whereas the effect was much less using CYTB probe (Figure 2A vs Figure S4A). Could the authors explain why? Or could they run a qPCR with a third probe?

- Pereira et al. showed mtDNA depletion in cybrids carrying 45% and 91% of m.8344A>G mutation respectively at day 1 and day 2. The observed effect is almost the same in these two clones. Is this effect due to the different time when the analysis was conducted? Since the mtDNA depletion is a consequence of the elimination of the mutant mtDNA after mitoTev-TALE transfection, should we expect a stronger depletion in cybrids harboring 91% of the mutation (as it has been suggested in the Discussion)?

- In Figure 2G, the authors claim that "the decrease in the mtDNA levels was not observed after 15-23 days after sorting". However, the results at 15*days are not clear. First, untreated cells should have a value of 100% because this sample should have been used as control to normalize all the others. I guess this value has been erroneously normalized to the untreated samples at 1 day. Second, the results showed a very big variability (considering the SEM calculated on 4-6 experiments) that makes it hard to draw any kind of conclusion.

Understanding that the mtDNA depletion is transient, it is extremely important to define whether the new approach represents a valid alternative to mitoTALENs. Therefore, these analyses need to be repeated at 15 and 25 days.

The OXPHOS function of cybrids with high levels of the mutation are reported in Figure 3. The data suggest that the transfection with mitoTev-TALE induces a shift in mtDNA heteroplasmy and a concomitant increase in oxygen consumption and in steady state level of NDUF8 and COXI.

- It would be better to represent in a single graph the results reported in Figure 3D and 3E (representing also the average of many experiments and not only one with the technical replicates).

- The figure 3F should also include the quantification of the black population.

- In Figure 3G two independent experiments/sorting have been shown. The results seem to be quite different. In the first one, as expected the rescue in the level of NDUF8 and COXI is stronger in the green population than in the black one. The second experiment presents an opposite scenario. More and new sorting/experiments and WB should be done to clarify this issue.

In the discussion the authors claim that "We showed that the heteroplasmy shift increases towards the WT mtDNA which was higher in the 45% mutant cybrids (25% increase in WT) when compared to the 11% increase observed in the 91% mutant cybrids, one or two days after transfection. Nevertheless, even in the highly mutant cybrids we were able to see a shift which was maintained up to 20-27 days after sorting. This could be explained by cybrid cell lines with a high mutation loads undergoing a more severe mtDNA depletion than those with a low mutant load". However, in this work the clone harboring 91% of mutation did not show the discussed more severe mtDNA depletion than the one harboring 45%. This point need to be better elucidated.

The manuscript needs to be substantially revised:

- in Figure 2D, some error bars are not visible (make smaller shapes)

- in all figures, the untreated population is named differently (UNT, Unt. and Unt.)

- Supplementary Figures contain really redundant results that can be pooled with data in the main figures (follow the comments above)

- The Figure S4 and S5 are missing any information regarding statistical analysis.

Reference

1. Bacman SR1, Williams SL, Pinto M, Peralta S, Moraes CT. Specific elimination of mutant mitochondrial genomes in patient-derived cells by mitoTALENs. *Nat Med* 2013

Referee #2 (Comments on Novelty/Model System):

This is a clearly written MS from a highly published lab regarding a novel technique MitoTalens, That corrects a mitochondrial defect, 8344A>G that causes the serious mitochondrial disease MERRF. So, the work is clearly done and convincing. There is potential medical impact given that they demonstrate MitoTalens-dependent correction of the defect at the genetic level. However there is a bit of a question regarding rescue at a functional level.

Referee #2 (Remarks):

This is an interesting MS regarding a novel variation on the Mito-Talens strategy upon which the Moraes lab has published 5 previous papers, called Mito-TeV, showing the correction of the 8344A>G mutation that causes MERRF. The MS is well and clearly written, and in general the figures are clear and concise and convincing, and support the idea that the Mito-TeV strategy helps with correction, reducing mutant DNA about 20% and thus relatively increasing WT mtDNA about 20%. There is some evidence of rescue of NDUFB8 expression and COXI by both Black and Green cells, and why this is is not well-explained, but there is rescue relative to untreated, which is good for their interpretation.

What is the hardest to interpret is the Seahorse data in Fig. 3 DEF, and its processing. First you should really say how many cells per Seahorse well were used, and whether this was a 24 or 96 well machine. In any event the respiration levels seem rather low when normalized to mg protein, seems like these should be in the 10-20pmol/min/ugprotein level. Also, the error bars are quite large and overlapping, suggesting not enough cells and not a significant rescue in respiration. Furthermore, after AA and Rotenone, O₂ consumption should be zero, and in figure 3 E it is not, so these two should be normalized to each other, to the height of that segment 7 of the graph, which would mean that the green and black are completely overlapping, i.e. no functional difference. So then with the overlap and NS results of D & E it makes it hard to understand all the significant differences of F. So, the MS is pretty convincing, that there is a genetic rescue of about 20% by Mito-TeV, but the functional rescue in Fig. 3DEF is not convincing, and it isn't clear the Seahorse experiments have been carried out or interpreted correctly. And some additional explanation about 3G why black rescues better than green in some cases would be appreciated.

Referee #3 (Comments on Novelty/Model System):

cybrid cells heteroplasmic for the m.8344A>G mutation (medium and high levels clones) are appropriate for these important proof of principle experiments.
this is a novel approach that - at present - isn't offering the highest medical impact but could certainly provide this in the future.

Referee #3 (Remarks):

The manuscript by Bacman and colleagues represents an extension of previously-published work from this group using mitochondrially-targeted TALENs to specifically shift mutant mtDNA levels towards wild-type in human cell culture (patient-derived cybrids) models.

The paper is well-written, articulate and presents evidence in support of this strategy in both medium level and high level m.8344A>G heteroplasmic cells, a well-characterised pathogenic mtDNA variant which causes MERRF syndrome. The extension of the TALEN-based therapy to a monomeric nuclease represents important work, and the data shown provide clear evidence that delivering enzymes to edit mtDNA shows therapeutic potential; the challenge of delivering this to clinically-relevant tissues remains but the work presented herein represents an important step towards this goal.

I only have a few comments and minor corrections as listed:

1. The quantitative data on assessment of mtDNA mutation load - key to the assessment of any technique aimed at shifting mtDNA heteroplasmy - are generated by last-hot-cycle PCR RFLP, a

long-established, tried and tested technique with alpha-32P label added in the last cycle of a PCR reaction so as to minimise quantitation errors due to heteroduplex formation. Quicker, fluorescent techniques are available yet the RFLP-assay is fine, provided experiments are performed (either in the design of the RFLP to engineer a further cut site for the enzyme or by spiking the sample with another labelled fragment which is cut the same enzyme) to ensure the restriction endonuclease cuts to completion. The slight concern I have about the data shown is that no such control is indicated - WT mtDNA is not cut therefore WT and uncut DNA is of the same length. Do the authors have data to show which confirms completion of digest? If not, it would be great to see the DNA samples assessed either with a re-designed RFLP or by another quantitative assay (e.g. pyrosequencing).

2. The experiments with the high level m.8344A>G mutation are prefaced in the manuscript with a reminder that the molecular mechanism (decreased efficiency of mt-tRNA-lysine aminoacylation) leads to a generalised disorder of mitochondrial translation. The authors have shown that the shift in m.8344A>G towards wild-type leads to improvements in cybrid cell respiration (Seahorse) and increased steady-state OXPHOS protein levels, but data on in vitro synthesis of mitochondrial proteins (35S translation) are not provided. I think this is really important, not only to show the biochemical defect in the cells being treated but further evidence that this process is restored and would really like to see these data added to the manuscript. A deeper assessment at a biochemical level would strengthen the conclusions; showing that the increased protein levels is associated with a restoration of the assembly of OXPHOS complexes (i.e. a simple one-dimensional BNPAGE) would likewise complement these results nicely - would the authors consider providing these data as well?

Minor comments:

1. line 11 of abstract, MERRF not MERFF
2. Introduction: to place the work in some broader context, it would be helpful to provide some details of the incidence of mitochondrial disease, particularly those due to mtDNA mutations.
3. Introduction: line 6: the authors comment that most mtDNA diseases have an early-onset; I'd suggest that this is probably not the case, perhaps changing to "variable-onset" better reflects the marked heterogeneity associated with these disorders in terms of genotype and clinical manifestation.

1st Revision - authors' response

17 May 2018

Answer to Reviewer's comments (in black)

General comments:

This manuscript was originally submitted approximately 11 months ago. Because of Reviewers requests, we repeated essentially all the experiments. As discussed below, the nomenclature of the sorted cells was changed. Because we have consistently identified fluorescent cells in cell populations initially sorted as "black" (i.e. GFP negative), we now refer to these populations as GFP-+.

Also because of the Reviewers' requests we performed several functional assays, showing a phenotype rescue in heteroplasmic MERRF cells by the mitoTev-TALE. We answered the specific points raised by the previous Reviewers below.

The text has changed substantially, so we did not highlighted the changes.

Referee #1 (Remarks):

Heteroplasmic mutations of mtDNA are common cause of mitochondrial diseases. Affected organs manifest the biochemical defect only when the proportion between mutated and wild type mtDNA exceeds a critical threshold. Recently, different approaches have been tried to reduce the copies of mutant mtDNA. For instance, Moraes' group developed a method to localize TALENs to mitochondria and cleave different classes of pathogenic mtDNA mutations leading to reduction in mtDNA heteroplasmy in patient-derived cells (1). The use of this approach has been suggested as

potential therapeutic tool for improving a patient's phenotype. Although promising, the system presents a crucial limitation in term of size of the enzymes (TALENs are dimers and large) that would make complicate to package their coding regions into a viral vector. In this manuscript the authors presented an attempt to facilitate the therapeutic delivery of mitochondrial editing enzyme in humans. Indeed, Pereira and colleagues described the new engineered monomeric mito Tev-TALE, as innovative strategy able to overcome the limitations in the architecture of mitoTALENs. As proof of concept, they showed that mito Tev-TALE can efficiently shift the mtDNA ratio towards WT copy in patient-derived cybrids harboring m.8344A>G mtDNA mutation.

Comments

The manuscript is well structured and shows a clear novelty in the field. However, I have many regarding the quality of the presented data. Therefore, I recommend a major revision.

- One of the main issues of this work resides on complete lack of consistency in the time points used for the analyses. For instance, cells are often analyzed after 1-2 days and between 15 and 23 days after FACS sorting. The second time point, described as 15* days, is actually a period of 8 days that most likely would explain the huge variations observed in many experiments. The authors should explain how and why they decided these time points and why these also differ between cybrids with 45% and 91% of mutation load. The experiments need to be performed again at 15 and 25 days.

After sorting cells, we faced the challenge of growing different cell populations, with different cell growth rates, simultaneously. Following the Reviewer's recommendations, we performed new sortings/experiments for all clones, and collect the cells at the same day still trying to maintain the cell passage as low as possible. Because of difference in cell growth after sorting, there is still a small time window that we had to group to do statistical analyses. However, these time points are much closer now.

- In Figure 2 and Figure S4, quantification of mtDNA levels after transfection with mito Tev-TALE were shown. The results seem to be not clear and convincing. In most of the experiments mtDNA-coded ND1 and CYTB were used as probes to quantify mtDNA (normalized for ACTB). I would suggest to plot these data on the same graph. At 1 day after transfection, the observed mtDNA depletion with ND1 as probe was quite severe whereas the effect was much less using CYTB probe (Figure 2A vs Figure S4A). Could the authors explain why? Or could they run a qPCR with a third probe?

We repeated the analyses of the old samples with another set of primers/probe specific for *COXI*. Now we show the analyses using *COXI* (Fig.S1D) and *ND1* (Fig.2E) for MERRF Low Mut clone (Clone 7). The results were essentially identical. For the subsequent experiments (with the High Mut clone, clone 20) we are showing only *COXI*, as it uses a more consistent Taqman method (Figure 3C and E).

Pereira et al. showed mtDNA depletion in cybrids carrying 45% and 91% of m.8344A>G mutation respectively at day 1 and day 2. The observed effect is almost the same in these two clones. Is this effect due to the different time when the analysis was conducted? Since the mtDNA depletion is a consequence of the elimination of the mutant mtDNA after mitoTev-TALE transfection, should we expect a stronger depletion in cybrids harboring 91% of the mutation (as it has been suggested in the Discussion)?

We have performed many more experiments to analyze mtDNA depletion. It is now clear that the MERRF mitoTev-TALE preferentially cleaves the MERRF mtDNA, but it can also cleave the wild-type mtDNA if the expression levels are high. We were able to show this concentration dependence for depletion and also for heteroplasmy shift. This concentration dependence is similar to what was previously described for mitoZFNs (Gammage et al., 2016). The revised manuscript discusses this issue in more details.

- In Figure 2G, the authors claim that "the decrease in the mtDNA levels was not observed after 15-23 days after sorting". However, the results at 15*days are not clear. First, untreated cells should have a value of 100% because this sample should have been used as control to normalize all the others. I guess this value has been erroneously normalized to the untreated samples at 1 day. Second, the results showed a very big variability (considering the SEM calculated on 4-6 experiments) that

makes it hard to draw any kind of conclusion.

Understanding that the mtDNA depletion is transient, it is extremely important to define whether the new approach represents a valid alternative to mitoTALENs. Therefore, these analyses need to be repeated at 15 and 25 days.

We have addressed this issue using the MERRF High Mut clone. We performed a new series of experiments with a more stringent gating during sorting. The results showed that the mtDNA depletion was dependent on the levels of expression. We discuss this potential concern in details in the revised Discussion.

The OXPHOS function of cybrids with high levels of the mutation are reported in Figure 3. The data suggest that the transfection with mitoTev-TALE induces a shift in mtDNA heteroplasmy and a concomitant increase in oxygen consumption and in steady state level of NDUFB8 and COXI.

- It would be better to represent in a single graph the results reported in Figure 3D and 3E (representing also the average of many experiments and not only one with the technical replicates).
- The figure 3F should also include the quantification of the black population.

We have performed several additional experiments to address the changes in phenotype, including new respiration assays. We have followed the reviewer's suggestion for the way we graph them. These experiments are now described in the new Figure 4.

- In Figure 3G two independent experiments/sorting have been shown. The results seem to be quite different. In the first one, as expected the rescue in the level of NDUFB8 and COXI is stronger in the green population than in the black one. The second experiment presents an opposite scenario. More and new sorting/experiments and WB should be done to clarify this issue.

More sortings/experiments were performed and combined with the previous results, for the same time-points. The cells were all cultured for up to 28 days, after sorting, when enough protein was obtained to perform the experiments. The new western-blot confirmed the improvement of the GFP-+ population which was in average as good as the improvement in the GFP++ population of cells (new Figure 4E).

In the discussion the authors claim that "We showed that the heteroplasmy shift increases towards the WT mtDNA which was higher in the 45% mutant cybrids (25% increase in WT) when compared to the 11% increase observed in the 91% mutant cybrids, one or two days after transfection. Nevertheless, even in the highly mutant cybrids we were able to see a shift which was maintained up to 20-27 days after sorting. This could be explained by cybrid cell lines with a high mutation loads undergoing a more severe mtDNA depletion than those with a low mutant load". However, in this work the clone harboring 91% of mutation did not show the discussed more severe mtDNA depletion than the one harboring 45%. This point need to be better elucidated.

Although there was a trend for a stronger depletion in the clone with higher levels of mutation (Fig. 2E vs 3C), there was quite a bit of variability on the different transfection/sorting results. Transfection efficiencies are also clone specific. Finally, we did show that at high concentrations the MERRF mitoTev-TALE can also cleave wt mtDNA, and this happens in a significant fraction of transiently transfected cells. WE expanded the discussion on this issue.

The manuscript needs to be substantially revised:

- in Figure 2D, some error bars are not visible (make smaller shapes)

All figures were extensively revised.

- in all figures, the untreated population is named differently (UNT, Unt. and Unt.)

This inconsistency has now been corrected.

- Supplementary Figures contain really redundant results that can be pooled with data in the main figures (follow the comments above)
- The Figure S4 and S5 are missing any information regarding statistical analysis.

We have pooled several Supplemental figures as suggested (Reduced to only 3 Supplemental Figures in the revised manuscript). Supplemental figures were also complemented with additional statistical and quantitative information.

Referee #2 (Comments on Novelty/Model System):

This is a clearly written MS from a highly published lab regarding a novel technique MitoTalens, That corrects a mitochondrial defect, 8344A>G that causes the serious mitochondrial disease MERRF. So, the work is clearly done and convincing. There is potential medical impact given that they demonstrate MitoTalens-dependent correction of the defect at the genetic level. However there is a bit of a question regarding rescue at a functional level.

Referee #2 (Remarks):

This is an interesting MS regarding a novel variation on the Mito-Talens strategy upon which the Moraes lab has published 5 previous papers, called Mito-TeV, showing the correction of the 8344A>G mutation that causes MERRF. The MS is well and clearly written, and in general the figures are clear and concise and convincing, and support the idea that the Mito-TeV strategy helps with correction, reducing mutant DNA about 20% and thus relatively increasing WT mtDNA about 20%. There is some evidence of rescue of NDUFB8 expression and COXI by both Black and Green cells, and why this is not well-explained, but there is rescue relative to untreated, which is good for their interpretation.

What is the hardest to interpret is the Seahorse data in Fig. 3 DEF, and its processing. First you should really say how many cells per Seahorse well were used, and whether this was a 24 or 96 well machine. In any event the respiration levels seem rather low when normalized to mg protein, seems like these should be in the 10-20pmol/min/ug protein level.

Also, the error bars are quite large and overlapping, suggesting not enough cells and not a significant rescue in respiration

The Seahorse experiments were conducted with 20 000 cells/well/100 μ l. We used a XFp Flux Analyzer, which has small wells (similar to the 96 well sizes). All this information is now incorporated in the Materials and Methods section of the paper. The experimental variability was within expected. The respiration in our rescued clones were lower than in the WT clone, but much higher than the untransfected mutant. We are now showing the SEM of n=5/7 different samples, coming from independent experiments/sortings, and each sample was analyzed at least in triplicate (Figure 4A).

Furthermore, after AA and Rotenone, O₂ consumption should be zero, and in figure 3 E it is not, so these two should be normalized to each other, to the height of that segment 7 of the graph, which would mean that the green and black are completely overlapping, i.e. no functional difference. So then with the overlap and NS results of D & E it makes it hard to understand all the significant differences of F.

The oxygen consumption after antimycin+rotenone is very low, but there is some non-mitochondrial oxygen consumption that is readily detectable. We have extended these experiments and now see a clear separation between untransfected and transfected cells.

So, the MS is pretty convincing, that there is a genetic rescue of about 20% by Mito-TeV, but the functional rescue in Fig. 3DEF is not convincing, and it isn't clear the Seahorse experiments have been carried out or interpreted correctly. And some additional explanation about why black rescues better than green in some cases would be appreciated.

This aspect has been greatly expanded in the new version of the paper. We now included additional Seahorse experiments, new western blots and mitochondrial protein synthesis (new Fig. 4).

Referee #3 (Comments on Novelty/Model System):

cybrid cells heteroplasmic for the m.8344A>G mutation (medium and high levels clones) are appropriate for these important proof of principle experiments.
this is a novel approach that - at present - isn't offering the highest medical impact but could certainly provide this in the future.

Referee #3 (Remarks):

The manuscript by Bacman and colleagues represents an extension of previously-published work from this group using mitochondrially-targeted TALENs to specifically shift mutant mtDNA levels towards wild-type in human cell culture (patient-derived cybrids) models.

The paper is well-written, articulate and presents evidence in support of this strategy in both medium level and high level m.8344A>G heteroplasmic cells, a well-characterized pathogenic mtDNA variant which causes MERRF syndrome. The extension of the TALEN-based therapy to a monomeric nuclease represents important work, and the data shown provide clear evidence that delivering enzymes to edit mtDNA shows therapeutic potential; the challenge of delivering this to clinically-relevant tissues remains but the work presented herein represents an important step towards this goal.

I only have a few comments and minor corrections as listed:

1. The quantitative data on assessment of mtDNA mutation load - key to the assessment of any technique aimed at shifting mtDNA heteroplasmy - are generated by last-hot-cycle PCR RFLP, a long-established, tried and tested technique with alpha-32P label added in the last cycle of a PCR reaction so as to minimize quantitation errors due to heteroduplex formation. Quicker, fluorescent techniques are available yet the RFLP-assay is fine, provided experiments are performed (either in the design of the RFLP to engineer a further cut site for the enzyme or by spiking the sample with another labelled fragment which is cut the same enzyme) to ensure the restriction endonuclease cuts to completion. The slight concern I have about the data shown is that no such control is indicated - WT mtDNA is not cut therefore WT and uncut DNA is of the same length. Do the authors have data to show which confirms completion of digest? If not, it would be great to see the DNA samples assessed either with a re-designed RFLP or by another quantitative assay (e.g. pyrosequencing).

An alternative RFLP strategy showing changes in heteroplasmy and complete digestion of the PCR fragment is now described in supplemental figure S2.

2. The experiments with the high level m.8344A>G mutation are prefaced in the manuscript with a reminder that the molecular mechanism (decreased efficiency of mt-tRNA-lysine aminoacylation) leads to a generalised disorder of mitochondrial translation. The authors have shown that the shift in m.8344A>G towards wild-type leads to improvements in cybrid cell respiration (Seahorse) and increased steady-state OXPHOS protein levels, but data on *in vitro* synthesis of mitochondrial proteins (35S translation) are not provided. I think this is really important, not only to show the biochemical defect in the cells being treated but further evidence that this process is restored and would really like to see these data added to the manuscript. A deeper assessment at a biochemical level would strengthen the conclusions; showing that the increased protein levels is associated with a restoration of the assembly of OXPHOS complexes (i.e. a simple one-dimensional BN-PAGE) would likewise complement these results nicely - would the authors consider providing these data as well?

We thank the reviewer for the suggestion and Figure 4D, has been now incorporated in the paper, showing an increased mitochondrial protein synthesis after transfection of Clone 20 with the MERRF mitoTev-TALE.

Due to the limited amount of samples obtained we did not perform the BN-PAGE, but we did perform additional western blots. We believe that now we have stronger evidence of improved mitochondrial function upon MERRF mitoTev-TALE transfection in a defective mutant cell clone.

Minor comments:

1. line 11 of abstract, MERRF not MERFF

This has been corrected.

2. Introduction: to place the work in some broader context, it would be helpful to provide some details of the incidence of mitochondrial disease, particularly those due to mtDNA mutations.

A sentence was incorporated in the introduction regarding this topic.

3. Introduction: line 6: the authors comment that most mtDNA diseases have an early-onset; I'd suggest that this is probably not the case, perhaps changing to "variable-onset" better reflects the marked heterogeneity associated with these disorders in terms of genotype and clinical manifestation.

The word was altered as requested.

Reference

Gammage PA, Gaude E, Van Haute L, Rebelo-Guiomar P, Jackson CB, Rorbach J, Pekalski ML, Robinson AJ, Charpentier M, Concordet JP et al (2016) Near-complete elimination of mutant mtDNA by iterative or dynamic dose-controlled treatment with mtZFNs. *Nucleic Acids Res* 44: 7804-7816

2nd Editorial Decision

15 June 2018

Thank you for the submission of your revised manuscript to EMBO Molecular Medicine. We have now received the enclosed report. As you will see, the reviewer is supportive and I am pleased to inform you that we will be able to accept your manuscript pending final editorial amendments.

I look forward to reading a new revised version of your manuscript as soon as possible.

***** Reviewer's comments *****

Referee #2 (Comments on Novelty/Model System for Author):

The authors have addressed all my concerns.

Referee #2 (Remarks for Author):

All my questions were addressed.

2nd Revision - authors' response

18 June 2018

Authors made the requested editorial changes.

Corresponding Author Name: Carlos T. Moraes

Journal Submitted to: EMBO Mol Med

Manuscript Number: EMM-2017-08084